# Chitosan-Based Scaffolds for Facilitated Endogenous Bone Re-Generation

**DOI:** 10.3390/ph15081023

**Published:** 2022-08-19

**Authors:** Yao Zhao, Sinuo Zhao, Zhengxin Ma, Chunmei Ding, Jingdi Chen, Jianshu Li

**Affiliations:** 1State Key Laboratory of Polymer Materials Engineering, College of Polymer Science and Engineering, Sichuan University, Chengdu 610065, China; 2Marine College, Shandong University, Weihai 264209, China; 3State Key Laboratory of Oral Diseases, West China Hospital of Stomatology, Sichuan University, Chengdu 610041, China; 4Med-X Center for Materials, Sichuan University, Chengdu 610041, China

**Keywords:** facilitated endogenous tissue engineering, chitosan, bioactive scaffold, functional design, bone repair

## Abstract

Facilitated endogenous tissue engineering, as a facile and effective strategy, is emerging for use in bone tissue regeneration. However, the development of bioactive scaffolds with excellent osteo-inductivity to recruit endogenous stem cells homing and differentiation towards lesion areas remains an urgent problem. Chitosan (CS), with versatile qualities including good biocompatibility, biodegradability, and tunable physicochemical and biological properties is undergoing vigorously development in the field of bone repair. Based on this, the review focus on recent advances in chitosan-based scaffolds for facilitated endogenous bone regeneration. Initially, we introduced and compared the facilitated endogenous tissue engineering with traditional tissue engineering. Subsequently, the various CS-based bone repair scaffolds and their fabrication methods were briefly explored. Furthermore, the functional design of CS-based scaffolds in bone endogenous regeneration including biomolecular loading, inorganic nanomaterials hybridization, and physical stimulation was highlighted and discussed. Finally, the major challenges and further research directions of CS-based scaffolds were also elaborated. We hope that this review will provide valuable reference for further bone repair research in the future.

## 1. Introduction

Bone tissue, as a dynamic living tissue, possesses a complex hierarchical structure. It plays an important function in body support, tolerance of force damage and the protection of internal vital organs, as well as providing a stable circulation environment for the bone marrow [1,2,3,4]. Normally, bone has a certain regenerative potential over time in the face of minor injuries [5,6]. In plenty of surgical trauma cases, however, once the critical-sized lesion is generated, it is difficult to achieve the effective repair of the defect tissue by the mere physiological regulation of an organism [7,8]. Clinically, exogenous implants are required to guide the repair of critical-sized bone defects. As the golden standard, auto/allogeneic transplantations have achieved positive effects, while limited donor sources, immunogenicity and the risk of infection caused by the second surgery restrict their widespread application [9]. Moreover, bone tissue engineering (BTE) consists of three important parts including a scaffold, cells, and growth factors, which achieves a good bone matrix simulation and overcomes the limitations of these transplantations mentioned above [10,11]. However, these traditional tissue engineering procedures are cumbersome, inconvenient, and time-consuming. Generally, tissue harvest, cell isolation, and ex vivo co-culture with a scaffold, as well as two invasive surgical procedures are included.

Inspired by an organism’s own repairing potential, facilitated endogenous bone tissue engineering (FEBTE) has been proposed as a more feasible approach to guide bone regeneration [12,13]. In comparation with traditional BTE, this strategy does not require the ex vivo culture of autologous cells and thus avoids the invasive surgical procedures with high risk [14]. In order to achieve the equivalent repair effect to BTE, a bioactive scaffold is usually employed in FEBTE strategy to in situ activate the intrinsic regenerative potential of native bone tissue and accelerate the tissue healing [15,16]. Therefore, the scaffold with appropriate material composition and bone-like structure is the key to recruit endogenous stem cells and growth factors to the damaged area in the organism. Some artificial metal/polymer implants (e.g., titanium and polymethyl methacrylate) were employed in this method. Nevertheless, the satisfactory host–material interface interactions and biosafety remain a challenge [17,18,19,20]. Subsequently, scientists have focused more on natural polymers such as bacterial cellulose, silk fibroin, and hyaluronic acid, etc., because many of them have good biocompatibility and non-immunogenicity to meet the requirements of in vivo transplantation [21,22,23]. However, these materials still have undeniable shortcomings; for example, pure cellulose shows a poor biodegradable ability in the physiological environment and poor osseointegration, which limits its further routine use in bone tissue engineering [24]. In addition, the weak mechanical properties and complicated purification process of hyaluronic acid, as well as the high production cost and limited source of silk fibroin, also weaken their use in bone tissue engineering (especially load-bearing bones) to a certain extent [25,26]. 

As a renewable source, chitosan (CS) is a naturally derived polysaccharide mainly produced from the exoskeleton of marine crustaceans [27], with a molecular structure and biological activity similar to the bone extracellular organic matrix [28]. The excellent biodegradability, biocompatibility and nontoxicity of CS have made it widely used in the field of bone repair [29,30,31]. Contrary to many synthetic materials such as polycaprolactone (PCL) and polylactic acid (PLA), the hydrophilic feature of CS can improve cell adhesion and growth on the scaffold surfaces [32,33]. Particularly, owing to the existence of a large number of amino and hydroxyl groups on the surface of CS, it is easy to be chemically modified and extensively designed [34,35,36,37]. For example, CS cross-linked with collagen (Col) showed better mechanical strength than pure Col scaffold, and the high porosity of CS/Col scaffold provided adequate space for the growth and differentiation of MC3T3-E1 cells [38]. Moreover, CS-based scaffolds can also act as a carrier to effectively control the release of osteo-inductive molecules, such as drugs [39,40], proteins [41,42], and peptides [43], etc., and then facilitate osteogenesis. Inspired by the biomineralization of natural bone, CS is often combined with other molecules to act as a mineralization template to induce the in situ crystallization of inorganic functional particles, such as bioactive hydroxyapatite (HAP) and magnetic ferric tetroxide (Fe_3_O_4_)[44]. Therefore, the versatile designability of CS-based scaffolds renders them promising candidates in the process of endogenous bone repair.

In this review, we mainly focus on the recent advances in chitosan-based scaffolds for facilitated endogenous bone regeneration (Figure 1). Initially, two bone repair strategies, FEBTE and BTE, are introduced and compared. Subsequently, CS sources, CS-based composite scaffolds and their fabrication techniques are briefly introduced. Furthermore, the functional designs of CS-based scaffolds in bone endogenous regeneration, including the loading of biomolecules, hybridization with inorganic nanomaterials and exogenous physical stimulation are highlighted and discussed in detail. Finally, the major challenges and further research directions of CS-based scaffolds are also elaborated. We hope that this review will provide valuable reference for further bone repair research in the future.

## 2. Bone Repair Strategies

### 2.1. Traditional Bone Tissue Engineering

Although auto/allogeneic transplantations, as the golden standard in the clinical setting, have achieved positive results in critical-sized bone defects repair, the limited donor sources, immunogenicity and the risk of infection caused by the second surgical procedure limit their widespread development [45,46]. In order to overcome the drawbacks of auto/allogeneic transplantations, the emergence of BTE has been warmly welcomed in the last three decades [47]. Briefly, the BTE strategy mainly consists of three parts: a scaffold, cells and growth factors [48]. As shown in Figure 2, the procedure begins with the isolation and harvest of target autologous tissues, then resuscitation and expansion of stem cells in specific culture equipment. After reaching a sufficient number, the cells are seeded into a prefabricated scaffold for co-cultivation in vitro. Simultaneously, suitable growth factors and nutrients are continuously added to provide beneficial conditions. Finally, the differentiated new tissues are implanted into the patient’s lesions to further interact with the host tissue, and finally promote defect tissue healing in vivo [49,50,51].

Generally speaking, the success of relevant clinical trials has validated the feasibility of this strategy. However, judging by the collapse of several well-known companies specializing in this field, this strategy is not cost-effective [52]. It is not hard to see that the actual operation process of tissue engineering is complex and costly. The complexity mainly lies in the co-culture of autologous cells and scaffold, as well as the usage of two invasive surgical procedures. In addition, the high cost mainly results from ex vivo culture media, sera, growth factors and the bioreactor, as well as the sterile and delicate culture environment. Furthermore, the quality of engineered products is also uneven.

### 2.2. Facilitated Endogenous Bone Tissue Engineering

The issues of complexity and high cost need to be addressed if BTE is to avoid becoming an expensive therapy available only to the wealthy. Specifically, we need to develop large-scale automated and replicable production systems as alternatives to labor-intensive production process. In view of the organism’s own repair potential, the biological microenvironment is used as a bioreactor to simplify the tedious process of BTE and finally achieve bone healing in situ. Therefore, the facilitated endogenous bone tissue engineering (FEBTE) strategy has emerged [12].

In comparison with the BTE strategy, the FEBTE strategy as a novel practical approach tries to eliminate time-consuming and costly tedious process: tissue harvest, cell isolation and ex vivo co-culture with a scaffold. Interestingly, this strategy only requires the implantation of a bioactive scaffold into the bone defect sites to induce the defect tissue repair by itself (Figure 2). The bioactive scaffold functions as a “gravitational field” to attract and positively recruit endogenous stem cells and growth factors to the damage site, and then promotes stem cells proliferation and osteogenic differentiation, thereby repairing bone defects [16]. Therefore, based on the convenience and cost-effectiveness of the FEBTE strategy, it has successfully attracted extensive attention in the field of scientific research [15,53]. Among them, the most important and key point is to construct a bone repair scaffold with excellent osteo-inductivity. 

## 3. CS-Based Bone Repair Scaffolds and Their Fabrication Methods

### 3.1. Source of CS and CS-Based Bone Repair Scaffolds

CS, as a renewable source, is mainly derived from chitin that is found in the shell structure of marine shrimps and crabs [54] (Figure 3). Seafood processing factories throw away countless crustacean shells every day, resulting in huge waste of resources and environmental pollution. Research has found that the content of chitin in these shells is as high as 30%. As a business opportunity, investors can make high-value utilization of these valuable wastes through chemical extraction processes [27,55]. According to statistics, the global chitosan market size is expected to reach 4.7 billion dollars in 2027 [27].

Usually, commercially available CS is mainly prepared by a two-step process. That is, chitin is extracted and purified from crustacean shells, and chitosan is subsequently obtained by alkaline deacetylation of a chitin molecule [27]. In the first process, crustacean shells need to go through four procedures to be converted into chitin: pretreatments (washing and drying); demineralization (acid treatment); deproteinization (alkali treatment); and decoloration (chemical washing). In the second process, chitin is converted to CS by a deacetylation process (alkali treatment): hydrolysis of the acetamide groups and the trans arrangement of the C-2/C-3 substituents in the sugar ring [56]. Finally, various degrees of deacetylation and molecular weights of pure chitosan are obtained by post-treatments [57]. In general, CS is insoluble in neutral or basic solutions due to its special molecular structure, but it can be dissolved in acidic aqueous solutions (pH < 6.5) by the protonation of NH_2_ moieties [58].

The protonated amino groups make CS positively charged, which can easily bind to many negatively charged molecules through electrostatic interactions, and show inherent bactericidal properties, thus rendering CS with different functions [59]. Due to its unique molecular structure and polysaccharide-based properties, CS possesses various physicochemical properties and superior biological activity including good biocompatibility, biodegradability, antibacterial, antitumor activity and antioxidation [60,61]. Consequently, CS has acquired enormous attention in various of fields such as the pharmaceutical industry [62], food packaging [63], and tissue engineering [64,65].

Based on the versatile and unique properties, the CS scaffold matrix has received significant interest in relation to bone regeneration. On the one hand, CS can be chemically modified with quaternization [36], carboxylation [35] and mercaptan [66], etc., so as to be effectively conjugated with other bioactive materials (Gelatin, Col and alginate, etc.) to achieve synergistic osteogenesis [38,67,68]. Furthermore, CS modified with phosphocreatine and carboxylic acid groups are often used as an organic template simulating the bone collagen matrix to induce crystallization of HAP in situ [69,70]. On the other hand, CS-based composite scaffolds can effectively control the release of osteo-inductive molecules (drug/protein/peptide/exosome/gene, etc.) such as chrysin [39], bone morphogenetic protein-2 (BMP-2) [71], parathyroid hormone (PTH) [72], etc., and thus promote osteogenesis. Moreover, CS-based hybrid scaffolds with the integration of inorganic nanoparticles including bioglass (BG) [73] and calcium phosphate (CaP) [74] can induce bone repair by releasing functional ions (such as Si^4+^, Mg^2+^and Sr^2+^). Moreover, CS can also be combined with some materials that accelerate bone repair through exogenous physical stimulation including light [75], electricity [76] and magnetism [77]. Therefore, the CS-based scaffolds with versatile designs can be effectively applied to FEMTE.

### 3.2. Fabrication Methods 

The standard for preparing bone scaffold is to simulate the native extracellular matrix (ECM) as much as possible, thereby providing a biomimetic microenvironment for cellular migration, proliferation and differentiation. After more than 30 years of development in tissue engineering, many techniques and devices have been developed to construct three-dimensional (3D) porous scaffolds, and each of them possesses its own unique advantages and disadvantages. This review takes several commonly used preparation methods as examples to make the following brief introduction (Table 1).

#### 3.2.1. Freeze-Drying

In 1909, Shackell pioneered the freeze-drying method, using many biological materials [89]. Since then, the method has been applied in many other fields such as the bio-pharmaceutical industry [90,91], food industry [92], biomedical engineering [93], etc. The bioactive scaffold in FEBTE should possess a 3D bone-like porous structure. The freeze-drying method can fabricate 3D porous scaffolds with high porosity and a pore size ranging from 20 μm to 400 μm [94]. A freeze dryer mainly consists of five parts: refrigeration system, control system, vacuum system, sample area and condenser [95]. The sample first needs to be frozen at a low temperature and then quickly transferred to a freeze dryer. Ice crystals in the sample are sublimated directly under vacuum dehydration. Eventually, the positions occupied by the ice crystals in the scaffold are naturally transformed into pores of different sizes [96]. Based on this, the preparation process is simple, and the porosity and pore diameter of the scaffold are easily controlled by the freeze-drying temperature. In order to endow the scaffold with a special structure, directional freezing drying technology has also been developed. According to a previous report, an oriented porous 3D CS/graphene oxide scaffold was designed using the directional freezing technology. The obtained scaffold with anisotropic pores could guide the alignment of MC3T3-E1 cells, and thus render the CS-based scaffold to achieve potential osteogenesis [79].

#### 3.2.2. Electrospinning

Electrospinning, which can also be said to be an improvement on electrospraying, employs electrostatic forces to create fiber networks from liquid polymer [32]. Videlicet, the machine usually demands a spinneret, high tension voltage field and collector [81]. As the polymer solution flows from the tip of the syringe, the tension applied by the high voltage twists it into a so-called Taylor cone [97]. Subsequently, the charged polymer solution accumulates on the charged collector in the form of filaments under the action of electrostatic repulsion, which is accompanied by the volatilization of solvent. Common collectors are a grounded metallic plate, cylinder or disc. If fiber scaffolds with different structures are expected, they can be effectively controlled by changing the structure of the receiver. For instance, a rotating drums collector can obtain aligned electrospun fibers. In addition, the morphology and physical properties of the electrospun scaffold can be adjusted by the parameters of polymer viscosity, the rotation speed of the collector and the distance of the syringe to the collector [98]. 

Therefore, electrospinning technology makes is easy to acquire scaffolds with different porosity, mechanical properties and oriented structures, meaning that it is widely studied in the field of tissue engineering [99]. Recently, the CS electrospun fibers combined with different polymers *(*PCL and PLA), as well as various bioactive nanoparticles (HAP and BG), have exhibited great potential in facilitating cell proliferation and differentiation. For example, to mimic the physicochemical structure of bone, related work fabricated a nanofibrous poly (vinyl alcohol)/CS/carbonated hydroxyapatite (PVA/CS/CHAP) scaffold via electrospinning technology. This scaffold could promote cell adhesion and growth and potentially be applied for bone regeneration [82].

#### 3.2.3. Three-Dimensional Printing

Three-dimensional printing, also called additive manufacturing, was first utilized by Massachusetts Institute of Technology (MIT) in the 1990s [100]. In recent years, it has been vigorously developed in all walks of life, especially in the field of bone tissue engineering [101]. It mainly blends the same or different materials to form a scaffold with a 3D structure through an automated layer-by-layer continuous processing process [102]. In order to design a scaffold that is similar to the native ECM of bone tissue, the 3D printing technology must be capable of fabricating precise scaffolds with controlled and interconnected pores, and high mechanical strength to enhance cellular activity [84,103]. For example, Zafeiris et al. printed a CS/HAP scaffold at a low temperature by controlling flow, infill and perimeter speed, which created a cell-friendly living environment and finally promoted cell adhesion and proliferation on its surface [104].

After continuous development, 3D printing technology has been updated with more rapid prototyping technologies, such as fused deposition modeling, selective laser sintering and stereolithography that can more elaborately design biomaterials [104]. Among them, the CAD-aided design technology can simulate the construction of the ideal scaffold structure and shape on the computer, and then quickly print the scaffolds according to this preset model [105,106]. For example, by using rapid prototyping technologies, Zhu et al. developed a poly(L-lactide)/CS scaffold; the novel scaffold not only improved the porous structure and mechanical properties, but also showed great potential to preserve the bioactivities and release rate of the biomolecules [107]. Li et al. also fabricated a poly(L-lactide)/CS/HAP hybrid scaffold through this method. The obtained scaffold with interconnected porous structure could facilitate the proliferation and differentiation of pre-osteoblastic cells [108]. In addition, in order to obtain a more accurate complex structural scaffold that can better match the host tissue, micro-computed tomography (Micro-CT)-assisted technology can directly simulate and reconstruct the clinical damaged bone model, and then print it through a 3D printer [109].

Therefore, the advantage of 3D printing technology is that complex materials with different structures can be obtained through precise design, especially in the face of diverse and complex bone defects in the clinical setting, which are difficult to achieve by other methods. However, 3D printing technology still has some drawbacks, such as the toxicity of adhesives if not completely treated, and the structural damage to materials caused by high temperature, etc. [110] Necessarily, a perfect post-processing process should be considered. In recent years, fortunately, Olhero’s team developed a series of sintered-free CS-based biphasic CaP bone scaffolds by robocasting suppressing sintering as a post-printing process [111]. The absence of sintering enabled the addition of biomolecules or functional nanoparticles to the extrudable inks, such as antibiotic levofloxacin and magnetic iron-doped HAP nanoparticles [112]. This enhanced fabrication technology endowed CS-based scaffolds with superior functions for cancer therapy by strong magnetic hyperthermia or bioactive drugs. To enhance the strength of the scaffold, the latest work presented by Torres et al. used this low temperature additive manufacturing technique to obtain a CS-based hybrid scaffold. By adjusting the ratio of silk fibroin in the CS/CaP complex, the printing scaffold with a macropore size of 300 μm showed 17 MPa compressive strength and 0.26 GPa Young’s modulus in a dry state [83].

#### 3.2.4. Sol-Gel Method

In 1846, French chemist J.J. Ebelmen discovered that orthosilicate could hydrolyze to form a gel in air, thus establishing the sol-gel chemistry [113]. The sol-gel method is a technology of hydrogel preparation under mild conditions. The basic reaction process includes solvation, hydrolysis reaction and polycondensation [114]. Initially, chemically active ingredients mix with raw materials to form a liquid phase, then the liquid phase turns to a stable transparent sol system through the hydrolysis and polymerization. The obtained sol slowly polymerizes into a 3D network structure. Finally, the required materials are obtained after drying or heat treatment. Recently, the sol-gel method has been widely used in the preparation of engineering scaffolds in the biomedical field [115]. For example, Ma et al. prepared a CS/polyvinyl alcohol (PVA)/nanoSiO_2_ composite scaffold, which was applied for bone engineering by the sol-gel method. The obtained scaffold has excellent mechanical properties and osteogenic differentiation ability [85]. Other related teams also employed this method to integrate CS with HAP [115], BG [86] and halloysite nanotubes [116] to form organic–inorganic hybrid hydrogels for bone endogenous regeneration.

To sum up, the sol-gel technology requires simple equipment, and its operation is convenient. Moreover, it has many advantages, such as low treatment temperature, good chemical uniformity of the precursor solution, easy control of the reaction process, etc. Inevitably, it also has some limitations, such as large drying shrinkage and difficulty being implemented for mass production [113].

#### 3.2.5. Others

In addition to the above commonly used methods, some special-purpose preparation methods are also practiced. The gas foaming method, as a facile technology, has been developed for some polymers such as PCL and PLA, the molding of which is mainly conducted by blasting the gas (such as CO_2_ gas) inside the stent under pressure, followed by freeze-drying [117]. In general, the resulting scaffolds have relatively large porosity, which can be higher than 90%. Moreover, the pore size of the scaffold obtained by this method is generally larger (500~1000 μm) than freeze-drying, which benefits the growth of cells [118]. However, the obtained scaffold often exhibits an uneven pore structure, and the connectivity between the pores is relatively poor. Furthermore, to obtain a more complex structure of the scaffold, the freeze-drying technique synergistic with other techniques would be a good idea. For this reason, gas foaming and microwave irradiation methods could be combined to yield super-porous CS/HAP hydrogel with interconnective pores [87]. In addition, the combination of freeze drying and porogen-leaching out methods could produce a CS-based composite scaffold with a distinct gradient of pore size, which plays a vital role in osteochondral repair [88].

## 4. Multifunctional Design of CS-Based Scaffolds in Bone Regenerations

Due to its unique molecular structure and polysaccharide-based characteristic, CS and its derivatives possess various physicochemical properties and superior biological effectiveness, including good biocompatibility, biodegradability, antibacterial activity, antitumor activity and antioxidation. Naturally, CS and its derivatives have acquired enormous attention in the fields of bone regeneration combined with other osteo-inductive materials. In this section, the functional design of CS-based scaffolds in bone endogenous regeneration is introduced and discussed, such as biomolecular loading (drugs/proteins/peptides/exosomes/genes); (Table 2) inorganic nanomaterials hybridization (CaP/BG/GO/GdPO_4_/SiO_2_); and physical stimulation (hyperthermia/magnetism/electricity/light).

### 4.1. CS-Based Scaffolds Integrate with Osteo-Inductive Molecules to Mediate Osteogenesis

#### 4.1.1. Drugs

Some small-molecule drugs such as chrysin [39], silibinin [119], simvastatin [120], etc., have positive effects in inducing osteogenesis by increasing the production of bone-associated proteins or regulating the polarization of macrophages from pro-inflammatory phenotype (M1) to anti-inflammatory phenotype (M2). From the perspective of practical applications, these drugs not only have significant therapeutic effects, but they also have a wide range of sources. Rational and effective use exerts their pharmacological value, which cannot be replaced by any other materials. Due to the unique biological effects of drug molecules, their controllable and precise delivery can effectively stimulate the differentiation and immune regulation of stem cells in the defect site [116,121,122]. Therefore, the incorporation of osteogenic inductive drugs into chitosan-based scaffolds is currently a hot research topic in FEBTE.

For example, icariin, which has good osteogenicity, was loaded into a HAP/carboxymethyl chitosan/poly(lactide-co-glycolide) (HAP/CMCS/PLGA) scaffold fabricated by the emulsion template method and freeze-drying technology (Figure 4). The icariin-loaded CS-based scaffold could effectively improve the adhesion, proliferation and differentiation of the osteoblast. After 12 weeks of transplantation in rat calvarial defects, the icariin-loaded scaffold finally achieved the repair and regeneration of bone defects [123]. In addition, ursolic acid (UA) shows good anti-inflammatory and osteo-inductivity. The incorporation of UA into mesoporous HAP and CS (MHAP-CS-UA) hybrid scaffolds endows the scaffolds with good osteogenic effects. In the MHAP-CS-UA micro-scaffold, the released UA could significantly upregulate the expression of osteogenic-related genes and proteins though promoting the M2-type polarization of macrophages, simultaneously inhibiting the polarization of macrophages to pro-inflammatory macrophages (M1 type) [122].

#### 4.1.2. Proteins/Peptides

As natural biomolecules, proteins are naturally secreted from organisms and possess special osteogenic functions [124]. In the process of bone defect repair, the formation of blood vessels, nerves, the osteogenic differentiation of stem cells and even the regulation of the M2-type polarization of immune cells are all essential factors [41]. Different proteins exert different biological effects for osteogenesis. For example, vascular endothelial growth factor (VEGF) can promote angiogenesis in the process of osteogenesis [125], while interferon-g (IFNg) facilitates neuronal growth [126] and platelet-derived growth factor-AA (PDGF-AA) accelerates oligodendrocyte specification [126]. Moreover, bone morphogenetic proteins (BMPs) including BMP-2, BMP-4 and BMP-7, etc., as a group of highly conserved homologous signaling proteins, play a vital role in embryogenesis, organogenesis, and cell proliferation and differentiation [71,127]. Recently, the loading of those bioactive components into CS-based scaffolds has achieved prominent effects in bone repair. 

In addition, the function of the protein is closely associated with its key amino acid sequences (peptides) [128]. The combination of these specific functional peptides in the CS-based scaffold through electrostatic interaction, covalent bonding, etc., shows a more efficient osteogenic effect than the above-mentioned proteins [129]. Meanwhile, the structure of peptides is relatively stable, and their storage and preparation are relatively cheap and easy [130,131]. For example, the widely used arginine-glycine-aspartic acid (RGD) peptide found in ECM adhesion proteins such as fibronectin and laminin, exhibits excellent osteogenic function in cooperation with CS-based scaffolds [132]. On this basis, CS-based scaffolds with the combination of multiple peptides such as RGD and FRHRNRKGY (HVP) peptide, extracted from human vitronectin, have also achieved obvious osteogenic effects [43]. Thus, it is suitable to deliver peptides in bone defect sites and facilitate endogenous tissue regeneration.

#### 4.1.3. Exosomes

Exosomes are mainly secreted from multivesicular bodies and widely exist in cell mediums. Their diameter is approximately in the range of 30 to 150 nm [133]. Related molecules such as nucleic acids, proteins, lipids, as well as metabolites in exosomes play an important role in communication between cells and executing biological functions [134]. The advantages of exosomes in mediating osteogenesis are obvious: they do not cause immune inflammation, have no tumorigenic risk and do not require engineering modifications [135]. Hence, exosomes would be good substitutes for the function of stem cells in tissue engineering. In addition, relevant studies have found that in the process of bone defect repair, the exosomes can be phagocytosed by target cells such as osteoprogenitors, endothelial cells and immune cells, and then participate in osteogenesis, angiogenesis and immune regulation [136,137]. Based on these, exosome-integrated CS-based scaffolds have been widely used in endogenous bone repair [138].

For example, Shen et al. incorporated dental pulp stem cell-derived exosomes (DPSC-Exo) into CS hydrogel, which could facilitate the repair of alveolar bone and treat the periodontitis [139]. Moreover, the related research team employed the coralline hydroxyapatite/silk fibroin/glycol chitosan/difunctionalized polyethylene glycol hydrogel as the carrier of human umbilical cord mesenchymal stem cells-derived exosome to study the defect bone repair in SD rats. In vitro and in vivo studies have found that the exosome-integrated CS-based scaffolds could effectively recruit stem cells and promote stem cell proliferation and osteogenic differentiation. Further, the expression of osteogenesis-related proteins in a hydrogel-exosome group was significantly higher than that in a hydrogel group, which finally mediated defect repair [140].

#### 4.1.4. Genes

In the last few years, gene-mediated bone therapy as a new and effective technology has received extensive research attention [141]. It usually delivers the gene sequences to the defect site, and then uses their special gene encoding ability to activate the specific osteogenic pathway to regulate the osteogenic differentiation of stem cells [142,143,144]. The common gene types mainly include RNA interference (RNAi); non-protein codding microRNA (miRNA); and messenger RNA (mRNA). Among them, mRNA can carry genetic information and direct the corresponding protein synthesis, while RNAi and miRNA mainly play the regulatory role in various cell functions [145].

For instance, considering that siRNAs can target casein kinase 2 interaction protein 1 (siCkip-1) and soluble VEGF receptor 1 (siFlt-1), Jia et al. incorporated two small siRNAs into a chitosan scaffold to promote new bone regeneration. The results showed that siRNA-modified chitosan scaffold could maintain a longer time to keep the function of siRNA. After loading with siCkip-1 and siFlt-1, siRNA-modified chitosan scaffold could simultaneously enhance the osteogenesis and angiogenesis, thus promoting new bone regeneration in vivo [146]. Meanwhile, this dual siRNA-loaded chitosan scaffold could still be used for other hard tissues regeneration such as dental regeneration [147].

**Table 2 pharmaceuticals-15-01023-t002:** The osteoinductive CS-based scaffolds integrate with different molecules.

Types	Molecules	Composite Matrix	Key Results	Ref.
Drugs	Icariin	Carboxymethyl CS/HAP/poly(lactide-co-glycolide)	Improved the adhesion, proliferation and differentiation of MC3T3-E1 and finally achieved the repair of bone defects.	[123]
	Ursolic acid	MHAP/CS	Upregulated the expression of osteogenic-related genes through promoting the M2-type polarization of macrophages.	[122]
	Chrysin	CS/carboxymethyl Cellulose/HAP	Stimulated cell proliferation and promoted osteoblast differentiation.	[39]
Proteins	VEGF, BMP-4	Gelatin/CS	Induced bone regeneration by angiogenesis and osteogenesis.	[125]
	BMP-2	PCL/carboxymethyl chitosan	Supported the proliferation, differentiation and ossification of hBMSCs.	[148]
	BMP-2, insulin-like growth factor-1	CS/gelatin	Significantly enhanced osteoblastic differentiation.	[149]
Peptides	FRHRNRKGY (HVP), GRGDSPK (RGD)	CS	Increased osteoblast adhesion, proliferation differentiation and calcium deposition.	[43]
	Parathyroid hormone-derived peptide	CS/HAP	Remarkably stimulated new bone formation in rabbit radial defects (size: 1.5 cm).	[72]
Exosomes	Pulp stem cell-derived exosomes (DPSC-Exo)	CS	Greatly facilitated the repair of alveolar bone and treated the periodontitis.	[139]
	Human umbilical cord mesenchymal stem cells-derived exosome	HAP/silk fibroin/glycol CS/polyethylene glycol	Effectively recruited stem cells, promoted their proliferation and osteogenic differentiation, and finally mediated bone repair.	[140]
	hMSCs-derived exosome	CS	Significantly increased osteogenic induction, promoted calvarial bone repair.	[138]
Genes	microRNA (siFlt-1+siCkip-1)	CS	Enhanced the osteogenesis and angiogenesis, finally promoted new bone regeneration in vivo.	[146]
	miR-24	CS/gelatin	Promoted osteogenic differentiation and skull defect regeneration in vivo.	[150]
	miR-590-5p	CS/HAP/nano-ZrO_2_	Upregulated osteogenic genes (RUNX2, COL I, ALP) expression and promoted osteoblast differentiation.	[151]

### 4.2. CS-Based Scaffolds Functionalized with Bioactive Nanomaterials to Induce Osteogenesis

#### 4.2.1. Calcium Phosphate

In order to improve the mechanical strength and bioactivity of CS scaffolds, it would be a good choice to hybridize bioactive inorganic nanomaterials to form organic–inorganic composites. In this regard, calcium phosphates (CaP) including HAP; amorphous calcium phosphate (ACP); dicalcium phosphate dihydrate (DCPD); octacalcium phosphate (OCP); and tricalcium phosphate (β-TCP) have been applied to bone regeneration, owing to their excellent osteo-inductivity [28,74,152,153,154]. Among them, the research on HAP nanoparticles with bone-like inorganic components is the most popular. For instance, Lou et al. mixed HAP nanoparticles with CS/gel composite to fabricate the hybrid composite scaffold by freeze-drying technology. It was found that the cell viability in the scaffold was enhanced with the increase in HAP content because HAP could promote cell adhesion and growth [155]. Furthermore, Huang et al. blended HAP nanoparticles into a CS and hyaluronic acid (HA) matrix to fabricate the CS/HA/HAP scaffold. They found that the addition of HAP nanoparticles effectively improved the mechanical stability of the CS/HA scaffold, and the rough surface of the scaffold was conducive to cell adhesion, proliferation and osteogenic differentiation [156]. To achieve FEBTE, a magnetic lanthanum-doped HAP/CS (MLaHAP/CS) scaffold was fabricated for the first time. The results found that the MLaHAP/CS scaffold could facilitate the osteogenic differentiation of rBMSCs by upregulating the phosphorylation of the Smad 1/5/9 pathway, and modulate the immune responses by regulating macrophage polarization into the M2 type to mediate osteogenesis [15].

However, HAP mixed directly with organic polymers might lead to the aggregation of inorganic nanoparticles into the polymer matrix, which can weaken the stability and bioactivity of the scaffolds [157]. In order to mimic the biomineralization process of natural bone tissue, biomimetic mineralization technology is gaining increasing interest among researchers [28]. Therein, a CS-based organic matrix as a template to regulate the growth of HAP has been correspondingly fabricated [158]. As shown in Figure 5, a summary of past research found that there are four mineralization methods to fabricate a CS-based HAP hybrid scaffold, including the wet chemical method [159,160], simulated body fluid [161,162], polymer-induced liquid precursor method [163,164] and alkaline phosphatase (ALP)-induced method [165]. These methods have their own characteristics. Among them, the preparation conditions of the ALP-induced method are relatively mild. In recent years, the research on mixing glycerophosphate and ALP to induce mineralization in hydrogels has gradually increased [165]. The phosphate groups in glycerophosphate turn to free phosphate groups under the action of ALP, and the calcium salt in the soaking solution is combined with free phosphate groups to form HAP nanoparticles [166].

In addition, polymer-induced liquid precursor method needs only one step to grow HAP nanoparticles in situ. For instance, Zhao et al. incorporated the precursor particles of HAP into the CS and GO covalent bonding network matrix to prepare a GO/CS/HAP hybrid scaffold via in situ one-step bionic technology (Figure 6). The in situ biomineralized scaffold overcame the drawbacks of HAP agglomeration when mixed directly with organic polymers, which further improved the bioactivity and osteo-inductivity of the composite scaffold. In vivo tests showed that this bioactive scaffold could in situ recruit endogenous stem cells to damage sites and promote endogenous bone tissue regeneration [16].

#### 4.2.2. Bioactive Glass 

Another inorganic biological material is bioactive glass (BG). As a promising candidate, it shows outstanding osteogenic activity and osseointegration between implants and native bone tissue [86]. BG with an amorphous structure can promote the proliferation and osteogenic differentiation of cells by the dissolved ions. Notably, among these functional ions, Mg^2+^, Sr^2+^, Si^4+^, Cu^2+^, Ca^2+^ and PO_4_^3−^ ions are most commonly used in facilitating angiogenesis, stem cell differentiation and immune regulation [73,167,168]. Therefore, doping these functional ions into BG particles to complex with a CS organic matrix would show great potential in the bone repair process [169].

For example, Wu et al. incorporated BG nanoparticles containing Ca^2+^, Si^4+^ and Cu^2+^ into a CS/silk fibroin/glycerophosphate (GP) composite to facilitate bone regeneration. Prepared hydrogel showed great bioactivity in vitro. Simultaneously, the controlled release of Si, Ca and Cu ions could effectively promote angiogenesis and osteogenesis in critical-size rat calvarial bone defect after 8 weeks of implantation. Furthermore, this cost-effective hydrogel with cell-free bioactivity shows great translation potential for endogenous bone regeneration [170]. Additionally, other BG nanoparticles doped with Sr and Mg ions still showed excellent mechanical stability and osteogenic potential after mixing with CS [167].

#### 4.2.3. Carbon-Based Nanomaterials 

Carbon, as an important element of biology, plays an indispensable role [171]. Carbon-based nanomaterials such as graphene oxide (GO) and carbon nanotubes (CNT) possess merits of large surface area and excellent mechanical strength, as well as good chemical stability and biocompatibility [172,173,174]. Therefore, they attract obvious interest for use in biomedical engineering.

Recently, Kaur and co-workers reported that carboxylated single wall carbon nanotubes (COOH-CNTs) show great biological advantages. They integrated COOH-CNTs into CS/Col hydrogels. The existence of COOH-CNTs obviously increased the mechanical stress of the hydrogels from kPa to MPa, similar to that of the bone. Furthermore, the hybrid hydrogels could adsorb HAP on their surface rapidly and, thus, enhanced the bioactivity of hydrogels [175]. In addition, to develop a bone scaffold with good osteo-inductivity, water uptake and retention, and mechanical properties, Ruan and co-workers fabricated a biocompatible scaffold through the chemical crosslinking of GO and carboxymethyl CS. The obtained CS-based scaffolds showed a higher water retention (44% water loss) than that of pure organic scaffolds (120% water loss). Furthermore, the mechanical property of the hybrid scaffold was at least 2.75-fold higher than that of the carboxymethyl CS scaffolds. Most importantly, the scaffold exhibited excellent regeneration effects in repairing SD rat calvarial defects [176]. In short, carbon-based nanomaterials can be used as bone substitutes for tissue regeneration.

#### 4.2.4. Gadolinium Orthophosphate

Rare earth element such as gadolinium (Gd), found in human bodies, plays an important role in accommodating cell differentiation, metabolism and tissue regeneration [177]. Recently, gadolinium orthophosphate (GdPO_4_) nanoparticle has attracted special interest for bone regeneration owing to it favorable biocompatibility and osteogenic potential [178]. The GdPO_4_ nanoparticles can release Gd^3+^ and PO_4_^3−^ after degradation, all of which can accelerate bone repair. For example, Zhao et al. prepared a GdPO_4_/CS scaffold by the freeze-drying method, as-released Gd^3+^ had non-toxicity to rBMSCs. The GdPO_4_/CTS scaffolds could significantly enhance the osteogenic differentiation of rBMSCs via the activated Smad/Runx2 signaling pathway, and finally mediate the collagen deposition and bone regeneration in the rat critical calvarial defect [179]. Subsequently, in the second year, their team blended Fe_3_O_4_ and GdPO_4_ nanoparticles into CS scaffolds. Under the NIR laser irradiation, this scaffold could effectively kill tumor cells by the photothermal effect, subsequently promoting osteogenesis by regulating the macrophages polarization of M2 type [180]. Hence, the composite material of GdPO_4_/CS exhibits a good prospect in the field of bone repair.

#### 4.2.5. Silica Minerals

In the field of biomedical engineering, silica minerals (SiO_2_)/CS hybrid scaffolds have shown great biological activity in bone regeneration by enhancing cells adhesion and growth, as well as promoting biomineralization [181,182]. In a related study, it was found that the Si-OH groups formed in SiO_2_/CS composite hydrogels could act as the inducer to promote HAP crystal nucleation in SBF, thus enhancing the bioactivity and osteo-conductivity of the hydrogel [183]. In addition, N-guanidinium-chitosan acetate/silica hybrid scaffolds containing either sulfonate or carboxylate groups could function as template to induce biomineralization [181]. At the same time, the scaffold prepared by the sol-gel transformation of SiO_2_ nanoparticles-PVA/CS mixed solution had good cyto-compatibility, which could significantly promote cell adhesion and growth [85]. Due to the unique structure and large comparative area, SiO_2_ nanoparticles could also be loaded on CS scaffolds as carriers for the delivery of drugs, proteins, etc., thereby enhancing the special osteogenic properties of the scaffolds [182].

### 4.3. CS-Based Scaffolds Synergize with Physical Stimulation to Promote Osteogenesis

#### 4.3.1. Hyperthermia Stimulation

In the last few years, hyperthermia therapy induced by light, electrical and magnetic stimulation has attracted extensive attention in biomedical fields [184,185,186]. In particular, the near-infrared (NIR) light in the wavelength of 700 to 1300 nm is utilized in the biomedical engineering field to trigger biological responses noninvasively [94]. NIR light stimulation shows deep tissue penetration as well as high spatial and temporal precision [187]. In the field of tissue engineering, a large number of scientific experiments have proved that the mechanism of heat-mediated repair is mainly the activation of heat stress-related pathways, and thereby the promotion of tissue repair [188,189]. According to the report, an MSCs membrane-coated black phosphorus (BP) photosensitizer was added into CS/Col hydrogel. Under the NIR irradiation, the MSC membrane-coated BP nanosheets in hydrogel could induce a mild photothermal effect to recruit the osteoblast via activating the heat shock proteins (HSPs)-mediated matrix metalloproteinase (MMP) and ERK-Wnt/β-catenin-RUNX2 axis. Furthermore, the thermal decomposition of BP could release PO_4_^3−^ to induce biomineralization. Finally, the BP-incorporated CS-based hydrogel could promote stem cells migration/differentiation and induce the biomineralization process to accelerate bone endogenous healing in the cranial defect of SD rats [190]. In view of the good biological effect of the photothermal effect, as shown in Figure 7, a temperature-controlled multifunctional HAP/GO/CS scaffold was also designed. In vitro and in vivo experiments showed that photothermal synergistic HAP/GO/CS scaffolds could damage osteosarcoma cells and simultaneously accelerate hard and soft tissue regeneration at the temperature of ~42 ℃ [191].

#### 4.3.2. Magnetic Stimulation

In recent years, magnetic stimulation has been a novel strategy to regulate cell behavior and mediate bone repair [192,193]. Magnetic fields can affect the alignment and growth of cells along the direction of the magnetic force, and benefit the stem cell proliferation and osteogenic differentiation, which are related to the NF-κB, integrin, MAPK and BMP pathways [194]. In vivo experiments demonstrated that the magnetic fields could remarkably enhance the integration between magnetic scaffold and host tissues, and finally facilitate bone regeneration [195]. For instance, ytterbium-doped hydroxyapatite (YbHAP) nanoparticles were in situ deposited in magnetic ferrites (SrFe_12_O_19_)/CS hybrid scaffolds. Both the magnetic fields and Yb^3+^ ions released from the hybrid scaffolds could activate the osteogenic-related BMP-2/Smad signal pathway, and simultaneously up-regulate the expression of VEGFs. The successful repair of rat calvarial defect further illustrated the promising potential of magnetic field in synergy with CS-based scaffolds to promote endogenous bone regeneration [196]. Interestingly, driven by the ambient magnetic field in the Earth, the in situ CS/Col/HAP/Fe_3_O_4_ magnetic scaffold prepared by Zhao and co-workers could in situ recruit endogenous stem cells and chemokines to damaged sites and facilitate osteogenic differentiation to achieve endogenous bone tissue regeneration, although there was no clear mechanism to explain it well [44].

#### 4.3.3. Electrical Stimulation

In the human body, the endogenic electrophysiological microenvironment plays an important role in maintaining the normal physiological functions and activities in ECM [197,198]. Particularly in load bearing bone tissue, the behaviors of cells such as biomineralization and cells differentiation, as well as cells polarization, are influenced by endogenic bioelectric cues [199]. To facilitate endogenous bone regeneration, an electrochemically responsive bioactive scaffold was prepared by integrating CNT and BMP-2 into carboxymethyl CS hydrogel. The conductive hydrogel responded sensitively to the differentiation degree of cells on both cellular and animal levels. Interestingly, the scaffold synergized with electrical stimulation could significantly accelerate new bone tissue formation into skull defects [174]. Therefore, the electrochemically responsive bioactive CS-based scaffolds will make an important step in mediating bone regeneration.

#### 4.3.4. Photobiomodulation

In recent years, photobiomodulation (PBM) has begun to acquire popularity in the clinical setting. In particular, visible light plays an important role in regulating osteogenesis. Compared with biochemical treatments, PBM might provide more biosafety and precise therapies for tissue regeneration and rehabilitation because of its high temporal, spatial accuracy and non-invasiveness. In biology, PBM therapies can achieve precise cell regulation by modulating specific signaling pathways [75]. Under green light irradiation (wavelength: 540 nm), the osteogenic-related gene expression levels of both SPP1 and BGLAP were significantly elevated, and finally promoted osteogenesis by activating the BGLAP and RUNX2 signaling pathway. At the same time, in contrast to the hyperthermia treatment mentioned above, this green light therapy can effectively avoid unnecessary tissue damage caused by high temperature [200]. Further research found that both green light (wavelength: 540 nm) and blue light (wavelength: 420 nm) could induce bone regeneration by regulating calcium concentrations in stem cells [201].

## 5. Conclusions and Prospects

In this review, we mainly compared and introduced two strategies, BTE and FEBTE, and then discussed the role and function of CS and its derivatives as a main component of bioactive scaffolds for facilitated endogenous bone regeneration. As a renewable source, many advantages of CS, including good biocompatibility, osteo-conductivity/osteo-inductivity, and biodegradability, make it increasingly popular in the field of bone repair. All of these CS-based scaffolds can be developed by freeze-drying, electrospinning, 3D printing, sol-gel or two combined approaches. Based on the versatile structure, CS can be chemically modified and functionalized with different bioactive materials (either organic molecular or inorganic nanoparticles) to achieve synergistic osteogenesis. In particular, the CS-based organic matrix can act as a template to regulate the growth of HAP in situ. Meanwhile, CS scaffolds can be used as controlled release platforms for the delivery of efficient osteo-inductive molecules and functional bioactive ions or combined with responsive materials to accelerate bone regeneration through exogenous physical stimulation. According to the positive results summarized in this review, the CS-based scaffolds with versatile design and functionalization can be effectively applied to FEBTE.

Although CS-based scaffolds have demonstrated unique therapeutic value in the field of bone repair, there are some challenges and prospects.

First, although the FEMTE strategy eliminates the tedious procedures of exogenous stem cells that traditional BTE relies on, making the bone repair process more convenient to operate, it is highly dependent on excellent bioactive scaffolds that can induce osteogenesis. Therefore, it is the key focus of future research to endow the scaffold with excellent osteogenic activity and exert its biological recruitment function of “gravitational field” in vivo.

Secondly, to further improve the osteo-inducibility of CS-based scaffolds, except for the compositional advantages discussed in this review, the structural design of scaffolds should be considered in the future. Related research reveals that different pore sizes have different effects on nutrient transport, cell, blood vessel and nerve growth, for instance, the optimal pore size for bone tissue in-growth is 200 to 350 μm [47]. In addition, the scaffold with the oriented pore channel has excellent anisotropic mechanical force and biological advantages, which can effectively regulate the growth behavior of cells [47,201]. For instance, Wang et al. designed a biomimetic bone scaffold with a honeycomb structure. The elastic modulus of scaffolds could match the elastic modulus of cancellous bone in the human body, and oriented channels could also accelerate cells to penetrate into the scaffolds, which was beneficial to bone repair [202]. Therefore, constructing CS-based scaffolds with a personalized channel or pore size to be compatible with the factors affecting bone repair will be a good direction to take in the future.

Third, the mechanical properties of CS-based scaffolds are relatively weak. Their special design from the perspective of biomimetic mineralization and 3D printing technology, so as to achieve a structure and function similar to natural bone, is expected to achieve its real clinical application. For example, inspired by the biomineralization process of bone, Chen et al. designed a 3D printed hydrogel with an enzyme-induced mineralization strategy. Through this endogenous mineralization process, the compression modulus of the obtained CaP-based mineralized scaffold (150 MPa) was much higher than that of the unmineralized hydrogel matrix (125 kPa). The hydrogel–mineral hybrid structures with unconventional tension-compression asymmetry show potential in bone reconstruction [166]. 

In conclusion, based on the discussion and outlook in this review, we believe that with the in-depth understanding of the mechanism of bone repair, the further optimization of repair scaffolds, and the rapid development of biomedical engineering science, future bone repair engineering will be more convenient and efficient.

## Figures and Tables

**Figure 1 pharmaceuticals-15-01023-f001:**
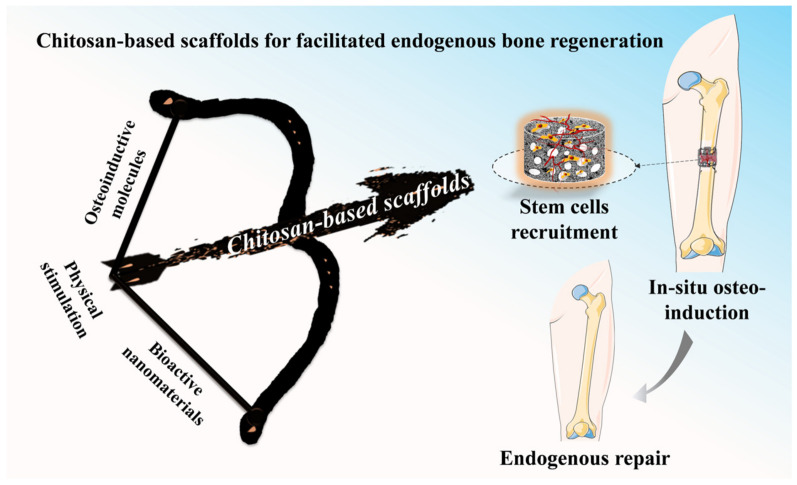
Multifunctional design of chitosan-based scaffolds and the application in facilitating endogenous bone regeneration.

**Figure 2 pharmaceuticals-15-01023-f002:**
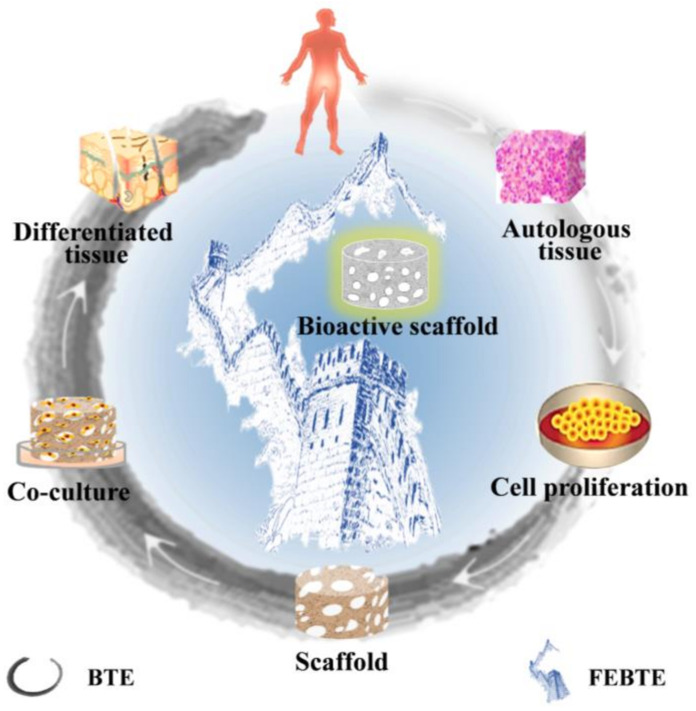
Comparison of two bone repair strategies. Traditional bone tissue engineering (BTE) needs tissue harvest, cell isolation and co-culture with a scaffold ex vivo, while facilitated endogenous bone tissue engineering (FEBTE) avoids these tedious and risky procedures by using a bioactive scaffold.

**Figure 3 pharmaceuticals-15-01023-f003:**
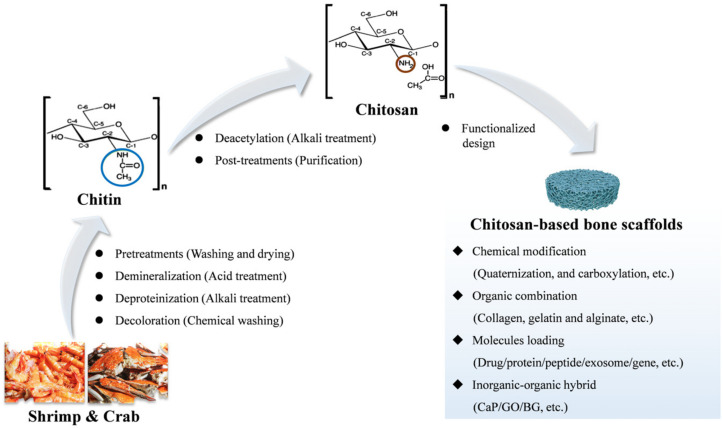
CS is extracted from crustacean shells and applied to the design of bone repair scaffolds through various functionalization strategies.

**Figure 4 pharmaceuticals-15-01023-f004:**
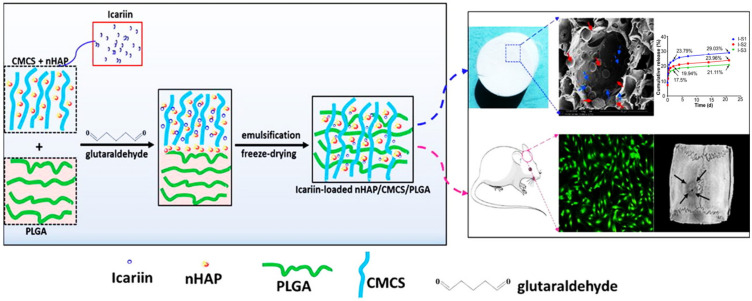
The preparation process of icariin-loaded HAP/CMCS/PLGA scaffolds and the application for cranial defects repair Reprinted with permission from ref. [123]. Copyright 2020 Chem. Eng. J.

**Figure 5 pharmaceuticals-15-01023-f005:**
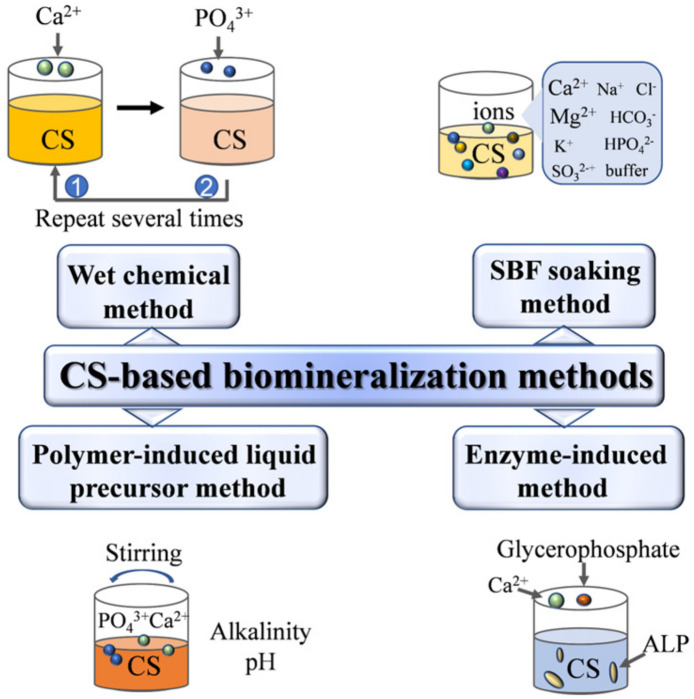
Biomineralization-inspired methods for the preparation of chitosan-based hybrid scaffolds.

**Figure 6 pharmaceuticals-15-01023-f006:**
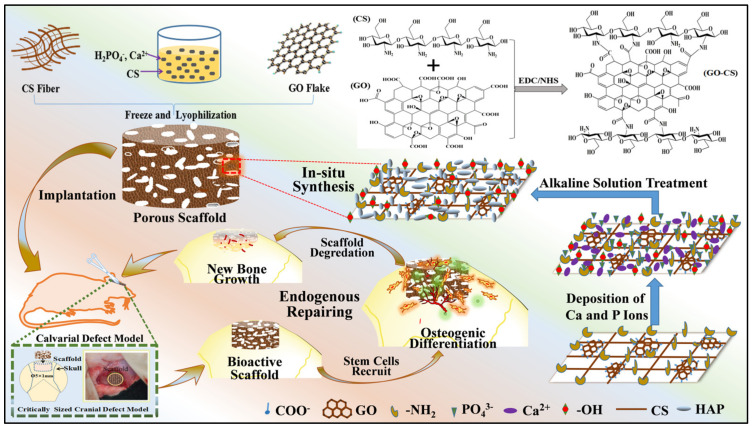
The GO/CS/HAP scaffold prepared by in situ mineralization strategy is used for endogenous bone regeneration. Reprinted with permission from ref. [16]. Copyright 2020 Chem. Eng. J.

**Figure 7 pharmaceuticals-15-01023-f007:**
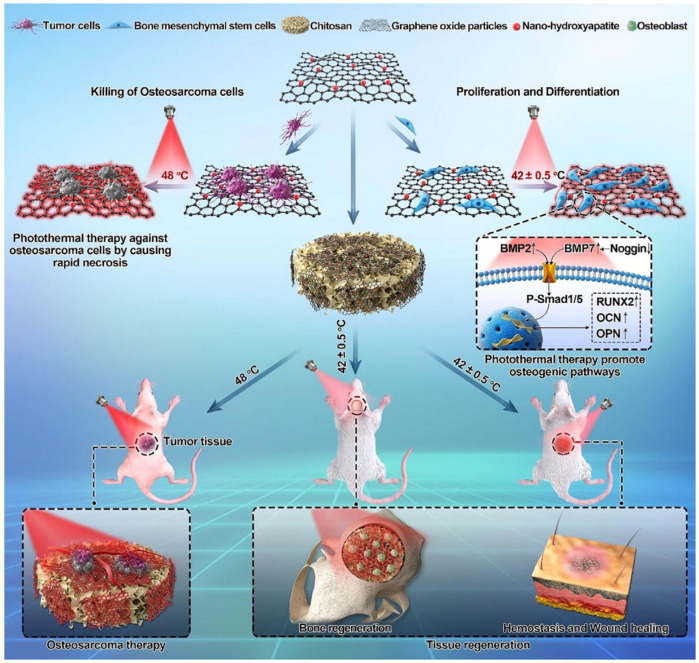
The photothermally controlled HAP/GO/CS scaffold for clinical treatment of osteosarcoma and tissue regeneration. Reprinted with permission from ref. [191]. Copyright 2020 Mater. Today.

**Table 1 pharmaceuticals-15-01023-t001:** The fabrication techniques and properties of CS-based scaffolds.

Fabrication Techniques	Composite	Important Properties	Ref.
Freeze drying	CS/graphene oxide/tetracycline hydrochloride	Controlled the drug release and promoted faster bone growth in rat femur defects.	[78]
	CS/graphene oxide	Oriented pores enhanced the alignment of MC3T3-E1 cells, facilitated osteogenesis.	[79]
Electrospinning	Zein/CS/polyurethane/carbon nanotubes	Facilitated cell proliferation, differentiation and upregulated the expression of osteogenic proteins.	[80]
	CS/HAP	Supported cell adhesion and promoted bone regeneration by activating integrin-BMP/Smad signaling pathway.	[81]
	CS/poly (vinyl alcohol)/carbonated hydroxyapatite	Promoted cell adhesion, growth and osteogenesis.	[82]
3D printing	CS/silk fibroin/cellulose	Osteo-immunomodulatory effects, accelerated bone regeneration in rat calvaria defects.	[61]
	Silk fibroin/CS/CaP	Enhanced the strength of scaffold, facilitated the proliferation and osteogenic differentiation.	[83]
	CS/HAP	Created a cell-friendly living environment, promoted cell adhesion, proliferation and osteogenesis.	[84]
Sol-gel method	CS/polyvinyl alcohol/SiO_2_	Excellent mechanical properties and osteogenic differentiation ability.	[85]
	CS/bioactive glass	Good shape memory properties and geometrical accommodation in bone implantation.	[86]
Gas foaming + microwave irradiation	CS/HAP	Scaffold with interconnective pores facilitated cells growth and upregulated osteogenic genes (RUNX2, OCN, COL I, ALP) expression.	[87]
Freeze drying + porogen-leaching out	CS/HAP	Scaffold with gradient pore and HAP composition implemented the bidirectional repair of osteochondral defects.	[88]

## Data Availability

Data sharing not applicable.

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
