# Peer review of "Chitosan-Based Scaffolds for Facilitated Endogenous Bone Re-Generation"

_pharmaceuticals, 2022, doi:10.3390/ph15081023_

Round 1
Reviewer 1 Report
In this submitted manuscript entitled "Chitosan-based scaffolds for facilitated endogenous bone regeneration”, the authors approach the topic in question in a carefully, well-structured and clear way. This manuscript reflects knowledge associated with the various approaches used to develop chitosan-based scaffolds with versatile design and functionalization by combination of bioactive materials and exogeneous physical stimulation in order to accelerate bone regeneration. However, there is a major concern about the statement in the field “CS-based bone repair scaffolds and their fabrication methods”. The authors should consider the recent works published regarding the production of composite scaffolds (chitosan and byphasic calcium phosphate based powders) by robocasting suppressing sintering as post printing process, such as (doi.org/10.1016/j.msec.2018.09.050, doi.org/10.1088/1748-605X/abac4c, and/or doi.org/10.1016/j.msec.2022.112690). These works seem to be interesting publications to be discussed in the manuscript and could enrich the review. I also advise that the authors write the acronyms in full, such as Gd (Gadolinium) and GO (graphene oxide) in the text. Therefore, I recommend this manuscript for publication in Pharmaceuticals MDPI.
Reviewer 2 Report
The review article entitled “Chitosan-based scaffolds for facilitated endogenous bone regeneration” have a systematic review of the application of Chitosan-based scaffolds for bone tissue engineering. The authors have also highlighted the various methods of preparation used for Chitosan-based scaffolds as well as the importance of CS-functionalized based bioactive scaffolds. The manuscript lacks a few important topics such as other natural polymers in bone tissue engineering with advantages and disadvantages compared to Chitosan, 3D printing for scaffolds. Thus, issues are needed to be addressed first before the recommendation of this review article for publication.
1. Authors need a section on natural polymers in bone tissue engineering with advantages and disadvantages
2. Explain how the Biological properties of chitosan make it possible to candidate of base materials for scaffold preparation in bone tissue engineering. Why chitosan is the best?
3. Authors need to describe Chitosan scaffolds made using rapid prototyping and 3D printing
4. The disadvantage of only Chitosan-based scaffolds needs to be more elaborated and methods to overcome it in bone tissue engineering have to be discussed with recent references.
5. There are many grammatical and sentence errors in the article, and the language organization needs to be improved.
Reviewer 3 Report
The manuscript reviews extensively the use of Chitosan-based scaffolds for endogenous bone regeneration. While the topic is very narrow, the authors have tried reviewing the field comprehensively. Few major suggestions:
1. Extensive editing of the language required
2. Some of the figures are not very professional. Figure 1- the arrow doesn't have any significance scientifically, looks more like an artwork; Figure 2- resolution very poor, FEBTE is not very clear and the building has nothing to do with the figure; Figure 6 - poor resolution.
3. Section 3.2.2 on 3D Printing only talks about the general steps involved and the 3D printing techniques. A few paragraphs about the works on 3D printed CS-based scaffolds will add value to the manuscript.
4. A comprehensive table on CS-based scaffolds fabricated using different techniques and their properties can be added.
5. Since the manuscript focuses on osteoinductivity, a table of osteoinductive CS-based scaffolds (drug loaded, protein, exosomes, etc that are discussed in section 4.1) and their results will really benefit the readers.
Round 2
Reviewer 3 Report
The manuscript still requires extensive language correction and improvement.